# Geldanamycin treatment does not result in anti-cancer activity in a preclinical model of orthotopic mesothelioma

M. Lizeth Orozco Morales[1,2,3], Catherine A. Rinaldi[1,2,4], Emma de Jong[5], Sally M. Lansley[3], Y. C. Gary Lee[3], Rachael M. Zemek[5], Anthony Bosco[5], Richard A. Lake[1,2,3], W. Joost Lesterhuis[1,2,3,5]*

1 School of Biomedical Sciences, University of Western Australia, Crawley, Western Australia, Australia, 2 National Centre for Asbestos Related Diseases, Nedlands, Western Australia, Australia, 3 Institute for Respiratory Health, Nedlands, Western Australia, Australia, 4 Centre for Microscopy Characterisation and Analysis, Nedlands, Western Australia, Australia, 5 Telethon Kids Institute, The University of Western Australia, Nedlands, Western Australia, Australia

* willem.lesterhuis@uwa.edu.au

**Data Availability Statement:** The generated datasets used in this manuscript have been deposited at Gene Expression Omnibus (GEO) and are publicly available. Accession number is

## Abstract

Mesothelioma is characterised by its aggressive invasive behaviour, affecting the surrounding tissues of the pleura or peritoneum. We compared an invasive pleural model with a non-invasive subcutaneous model of mesothelioma and performed transcriptomic analyses on the tumour samples. Invasive pleural tumours were characterised by a transcriptomic signature enriched for genes associated with MEF2C and MYOCD signaling, muscle differentiation and myogenesis. Further analysis using the CMap and LINCS databases identified geldanamycin as a potential antagonist of this signature, so we evaluated its potential *in vitro* and *in vivo*. Nanomolar concentrations of geldanamycin significantly reduced cell growth, invasion, and migration *in vitro*. However, administration of geldanamycin *in vivo* did not result in significant anti-cancer activity. Our findings show that myogenesis and muscle differentiation pathways are upregulated in pleural mesothelioma which may be related to the invasive behaviour. However, geldanamycin as a single agent does not appear to be a viable treatment for mesothelioma.

## Introduction

Mesothelioma is a cancer that usually arises in the pleura, occasionally developing from other serous membranes [1]. Its development is associated with asbestos exposure, with a latency period of approximately 40 years between exposure and diagnosis [2]. Mesothelioma's morbidity is predominantly caused by local invasion into neighbouring tissues such as lungs, heart, diaphragm, and chest wall [3]. On a cellular level, local invasion is a coordinated process that requires the cancerous cells to interact with the tumour microenvironment, the cell matrix, or other cells as part of a cell-cell adhesion process [4], with several signalling pathways controlling these interactions, as well as the cytoskeletal dynamics in the tumour cells and the cell movement into adjacent tissues [5].

GSE180618. The rest of the relevant data are within the paper and its Supporting information files.

**Funding:** This research was supported in part by grants by Cancel Council Western Australia (grant number APP1141445, https://cancerwa.asn.au/), iCare Dust Diseases Care (https://www.icare.nsw.gov.au/), National Health and Medical Research Council's (grant number APP1196605, https://www.nhmrc.gov.au/), and Simon Lee Foundation (https://simonleefoundation.com.au/) assigned to W.J.L. The funders had no role in study design, data collection and analysis, decision to publish, or preparation of the manuscript.

**Competing interests:** The authors have declared that no competing interests exist.

For many mesothelioma patients, palliative chemotherapy with cisplatin/pemetrexed is the first-line therapy [6]. Recently, cancer immunotherapy with nivolumab plus ipilimumab improved overall survival in mesothelioma patients compared to cisplatin/pemetrexed [7]. Combination immune-chemotherapy with anti-PD-L1 antibody durvalumab and cisplatin/pemetrexed showed a promising progression-free survival [8], which is now further explored in a randomized phase 3 trial [9]. However, despite this progress, most patients still do not respond to these treatments.

Recent developments in systems biology have allowed gene expression profiling to be used not only for comparing specific pathology but also to unravel drug-disease associations [10]. The connectivity map (CMap) dataset has been used as a systematic approach to connect gene expression profiles associated with disease with drug-induced expression profiles, and thereby identify drug repurposing candidates that are predicted to reverse or reinforce genomic signatures of disease [11]. In addition, the library of integrated network-based signatures (LINCS) L1000 dataset, has over a million gene expression profiles from cell lines treated with small molecules, growth factors, cytokines and drugs. This allows potential identification of drugs that are predicted to mimic or reverse the input gene expression signature by comparing the LINCS L1000 datasets and disease-specific signatures [12]. The expectation of this approach is that existing drugs with known safety profiles can be repurposed for alternative indications, accelerating the clinical development pathway [13].

Here, using transcriptomic data from invasive and non-invasive models of murine mesothelioma, we aimed to map an invasive signature of mesothelioma and identify drug repurposing candidates with potential anti-mesothelioma activity.

## Results

### Muscle development and myogenesis signatures are associated with the invasive pleural mesothelioma model

By inoculating the same mesothelioma cell lines AB1 and AE17 in tandem, we have previously shown that mesothelioma cells grow significantly faster in the pleural space compared to the subcutaneous space [14]. Additionally, we found that the intrapleural (IPL) tumour microenvironment induces or permits an invasive phenotype whereas the subcutaneous (SC) environment does not. We performed RNA sequencing of tumours from both the pleural and subcutaneous locations, while making sure only tumour was excised, not any surrounding normal tissue such as lung, heart, bone, or muscle (Fig 1A) [14]. Differential expression analysis [15] identified 419 genes that were differentially expressed between the IPL and SC tumours in both AE17 (C57BL/6 background) and AB1 (BALB/c background) mesotheliomas. Ingenuity Pathway Analysis [16] of the differentially expressed genes showed the two most significant predicted upstream transcription factors as myocyte-specific enhancer factor 2C (MEF2C), which maintains the differentiated state of muscle cells during myogenesis [17] (p. value = 3.4 x $10^{-8}$, z-score = 2.78) and myocardin (MYOCD), a co-transcriptional activator of serum response factor that inhibits the cell cycle and induces smooth muscle differentiation [18] (p. value = 1.5 x $10^{-5}$, z-score = 2.96).

To determine which biological pathways were upregulated in the IPL tumours, we used InnateDB [19]. This identified ten significant pathways from the Reactome and KEGG databases in both tumour models (p. value < 0.05) (Fig 1B). The most significant pathways were related to muscle contraction, cardiomyopathy and myogenesis. To determine the functional characteristics of the pleural invasion-related set of genes, we performed pre-ranked gene set enrichment analysis (GSEA) [20, 21]. This identified nine gene sets from the gene ontology

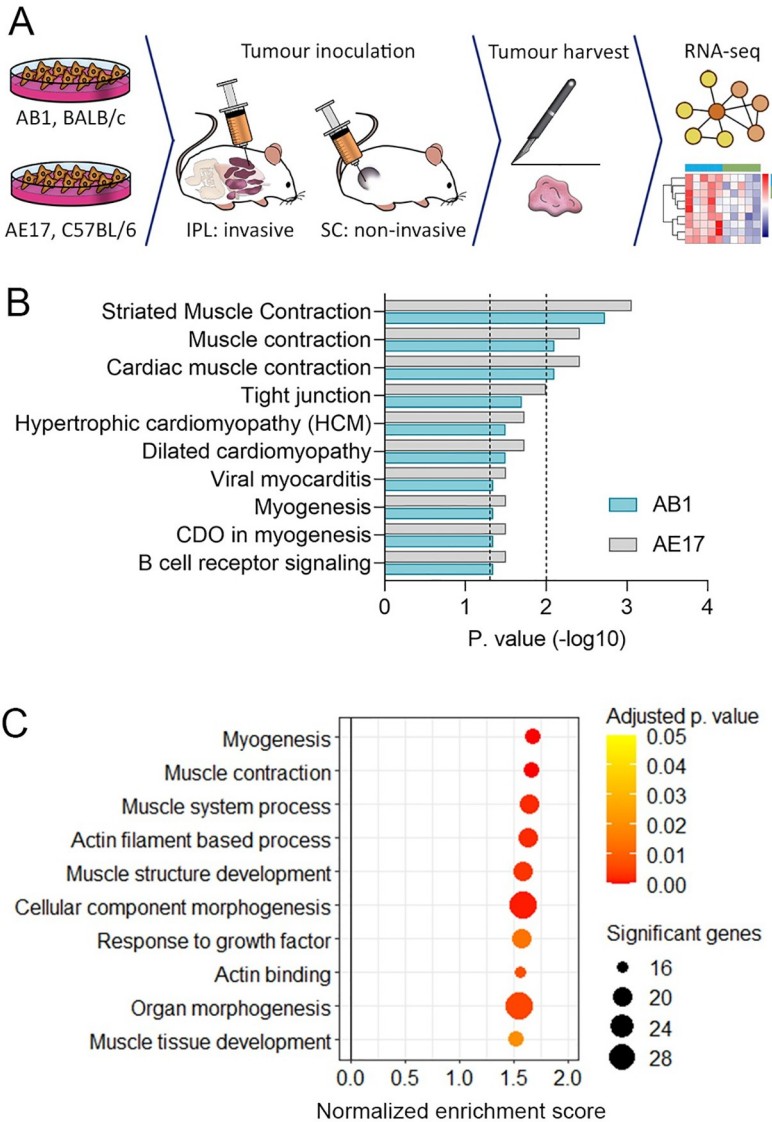

**Fig 1. Muscle development and myogenesis signatures are associated with the invasive pleural mesothelioma model.** (A) Experimental design, n (mice) = 8 per group. (B) Canonical pathways analysis of differentially expressed genes using InnateDB, p values 0.05 and 0.01 are represented with dotted line on–log10 = 1.3 and 2, respectively. (C) GSEA of differentially expressed genes, between the pleural and subcutaneous tumours in both AB1 and AE17. Circle size represents number of significant genes, colour represents adjusted p value.

biological process consortium [22], and one gene set from the hallmark collection [23], that were consistently enriched in both invasive models (p. value < 0.05, false discovery rate (FDR) < 0.25, Fig 1C). These gene sets mainly related to muscle development, morphogenesis, and actin processes, and myogenesis, suggesting that muscle development signatures were related to the invasive phenotype of pleural mesothelioma. Further analysis with the Descartes dataset in Enrichr [24–26] of the IPL differentially expressed genes (log fold change >1) while excluding immunoglobulin-related genes resulted in a mesothelial cell signature (S1A Fig), suggesting that environmental cues from the pleural space drive this mesothelial program more than the subcutaneous space.

## LINCS and CMap analyses identify geldanamycin as a potential inhibitor of mesothelioma invasion-related pathways

We went on to identify drug repurposing candidates that were predicted to target these mesothelioma invasion-associated pathways. We interrogated the LINCS L1000 Chem Pert repository [22] and the CMap database [11], accessed via the Enrichr platform [24–26], using the genes that were upregulated in the invasive (pleural) AB1 and AE17 mesothelioma models as input data. This analysis identified geldanamycin as the only drug to be significantly associated with the invasive signature in both databases (LINCS L1000; p value < 0.0001, CMAP; p. value = 0.012, S1 File). Geldanamycin and its derivatives are inhibitors of heat-shock protein 90 [27, 28], and previous studies demonstrated significant inhibitory effects on myogenic differentiation and muscle regeneration [29, 30], as well as direct anti-cancer effects, including in mesothelioma *in vitro* [31, 32]. Together, these results suggested that geldanamycin might have anti-cancer activity in pleural mesothelioma.

## Low-nanomolar concentrations of geldanamycin inhibit mesothelioma cell growth, invasion, and migration *in vitro*

Having identified geldanamycin as a possible drug to target mesothelioma invasion, we tested its effect on cellular proliferation in the murine mesothelioma cell lines AB1, AE17, the human mesothelioma cell lines VGE62, JU77 and MSTO-211H, and the non-cancerous fibroblast murine cell line NIH3T3. This showed $IC_{50}$ values at low-nanomolar concentration for all cell lines (Fig 2A–2F). In addition, we observed that geldanamycin significantly decreased JU77 growth in a soft agar colony formation assay at 6.25 nM concentration and higher (p. value < 0.0009; Fig 2G and 2H).

To evaluate the effect of geldanamycin on mesothelioma migration and invasion *in vitro* we employed scratch assays. Geldanamycin significantly inhibited migration (Fig 2I–2L) and invasion in a concentration-dependent manner (Fig 2M–2P).

## Geldanamycin treatment does not have anti-mesothelioma activity *in vivo*

To assess any therapeutic effect of geldanamycin on mesothelioma invasion and growth *in vivo*, we used our optimised model of orthotopic invasive mesothelioma [14] with AB1 cells expressing luciferase (AB1-Luc). The luciferase transduction had not changed the phenotype and geldanamycin sensitivity of the AB1 cell line, in terms of its expression of PD-L1, MHC-I and podoplanin, nor its *in vitro* invasive potential (S2A–S2G Fig). In addition, the basal levels of HSP90 did not change between cell lines, with or without geldanamycin (S2H Fig). We inoculated AB1-Luc intraperitoneally and administered geldanamycin twice daily for 5 days and monitored mesothelioma growth using bioluminescence imaging (Fig 3A). Optimization experiments showed that geldanamycin at a dose of 0.1 mg/kg was tolerable and did not result in any discomfort (Fig 3B, S3A and S3B Fig). This dose has previously been shown to result in a significant biological effect in vivo, in an oedema mouse model [33]. There was no significant effect on tumour growth in the geldanamycin group when compared to vehicle controls (Fig 3C and 3D). To determine whether geldanamycin had any effect on mesothelioma invasion, we collected the tumours and surrounding tissues for histological analysis. We found that geldanamycin did not affect mesotheliomas invasion, with both groups showing clear invasion of mesothelioma cells into the surrounding organs such as liver, intestine and pancreas (Fig 3E). Together, these data show that geldanamycin does not have any significant effect on the proliferation or invasion of mesothelioma *in vivo*.

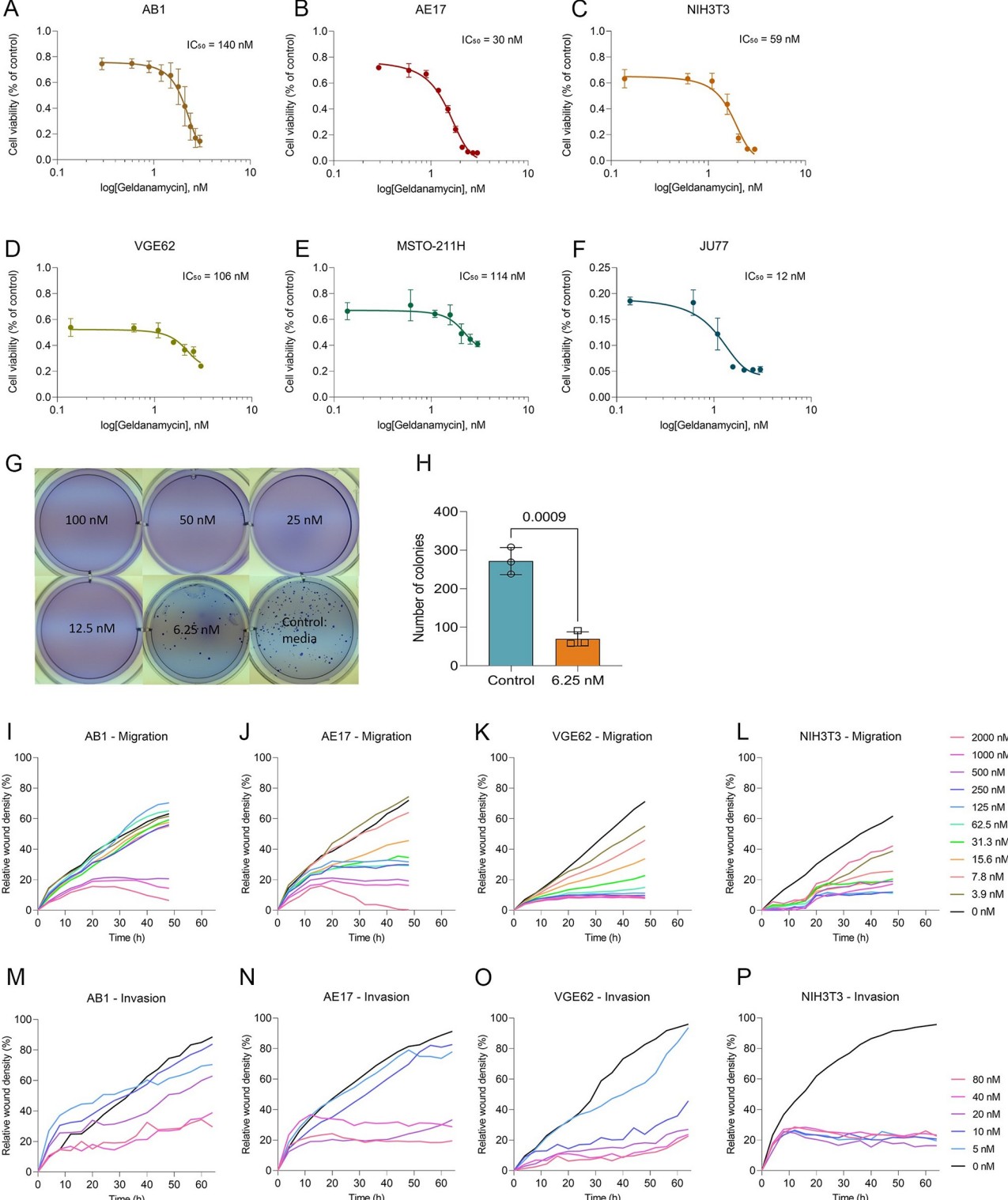

**Fig 2. Nanomolar concentrations of geldanamycin inhibit cancer cell growth, invasion, and migration *in vitro*.** (A to F) MTT assay in (A) AB1, (B) AE17, (C) NIH3T3, (D) VGE62, (E) MSTO-211H, and (F) JU77 cell lines. The curve is presented as a non-linear regression; log(geldanamycin) versus response. IC50 values are in nM. Data are presented as means ± SD of 3 replicates. (G and H); data are means ± SD of 3 replicates, unpaired t test was used to determine significance. Soft agar colony formation assay for JU77 human mesothelioma cell line. Geldanamycin was added at 100, 50, 25, 12.5, 6.25 nM and the control (0 nM). (G) Colony counting was performed using ImageJ software. Edges were excluded, size = pixel² 0 –infinity,

circularity = 0.00–1.00. (I to L) Scratch assay for migration in (I) AB1, (J) AE17, (K) VGE62, and (L) NIH3T3 cell lines. Geldanamycin was serial diluted from 2000 nM to 3.9 nM, with 0 nM as control. Data are means of 3 replicates. (M to P) Scratch assay with matrigel at 8 mg/mL for invasion in (M) AB1, (N) AE17, (O) VGE62, and (P) NIH3T3 cell lines. Geldanamycin was serial diluted from 80 nM to 5 nM and 0 nM as control. Data are means of 3 replicates.

## Discussion

Targeting invasion in the treatment of mesothelioma is of great interest, given the central role of invasion in the morbidity of this disease. Previous clinical studies have attempted to inhibit invasion in mesothelioma patients by targeting mesothelin [34, 35], a membrane-bound protein that stimulates anchorage-independent growth, migration, and invasion [36, 37]. However, progression free survival was not different between the treatment and control groups. Another study [38] targeted the Met signalling pathway, a known mesothelioma invasion promoting pathway [39]. However, no results have been reported to date.

In the present study, we aimed to identify regulators of mesothelioma invasion by comparing invasive intrapleural tumours with non-invasive subcutaneous tumours. Using transcriptomic data from these models, we identified MEF2C and MYOCD as upstream regulators of the invasive mesothelioma model. Altered MEF2C regulation has been implicated as driver of cancer development [40], and it acts as an oncogene for immature T-cell acute lymphoblastic leukaemia [41], myeloid leukaemia [42], hepatocellular carcinoma [43], and pancreatic ductal adenocarcinoma [44]. Perturbations in MYOCD, are associated with heart failure, acute vessel disease, diabetes and cancer [45], with some MYOCD-related transcription factors regulating cytoskeletal dynamics [46] which can benefit tumour migration and invasion [47].

Using canonical pathway analysis and GSEA, we identified muscle development and myogenesis signatures associated with the invasive model. Using the LINCS L1000 Chem Pert repository and the CMap database, we identified geldanamycin, an antibiotic isolated from *Streptomyces hygroscopicus* [48] and a naturally occurring benzoquinone ansamycin that targets Hsp90 [27], as a potential drug to target these myogenesis and muscle differentiation signatures in invasive mesothelioma. In our study, we observed that addition of geldanamycin did not affect the expression of HSP90 itself. These data agree with others that have shown that geldanamycin treatment does not downregulate HSP90 protein expression levels as such [49, 50], but rather inhibits HSP90's downstream targets [49, 50]. Another explanation for geldanamycin's mechanism of action as observed in our in vitro experiments is that it is HSP90-independent. Geldanamycin has previously been shown to inhibit muscle regeneration and myogenic differentiation [29, 30], and geldanamycin and its derivatives have been previously recognised as anti-cancer agents [51]. *In vitro* cancer studies in squamous cell carcinoma demonstrated significant inhibition of cell proliferation and G2 arrest by geldanamycin [52], and it inhibited angiogenesis and invasion in a prostate cancer model, mediated by the hypoxia-inducible factor 1α [53]. Particularly for mesothelioma, a study by Okamoto et al. (2008) showed that *in vitro* use of tanespimycin, a less-toxic derivative of geldanamycin, led to significant G1 and G2/M cell cycle arrest and apoptosis, while inhibiting cell proliferation in mesothelioma cells [54].

In our study, we also observed a significant reduction on tumour growth, cell migration and cell invasion *in vitro*, with an $IC_{50}$ in the nanomolar range for all the tested cell lines. However, the metabolism and cell movement of the non-cancerous cell line NIH3T3 was significantly inhibited as well, with an $IC_{50}$ of 59 nM. This non-cancer selective inhibition could be related to the toxicity that we observed *in vivo* when dosing geldanamycin at 1 and 0.5 mg/kg. Most importantly, we did not observe any anti-mesothelioma activity for geldanamycin in an orthotopic model, when given at a tolerable yet biologically active dose [33].

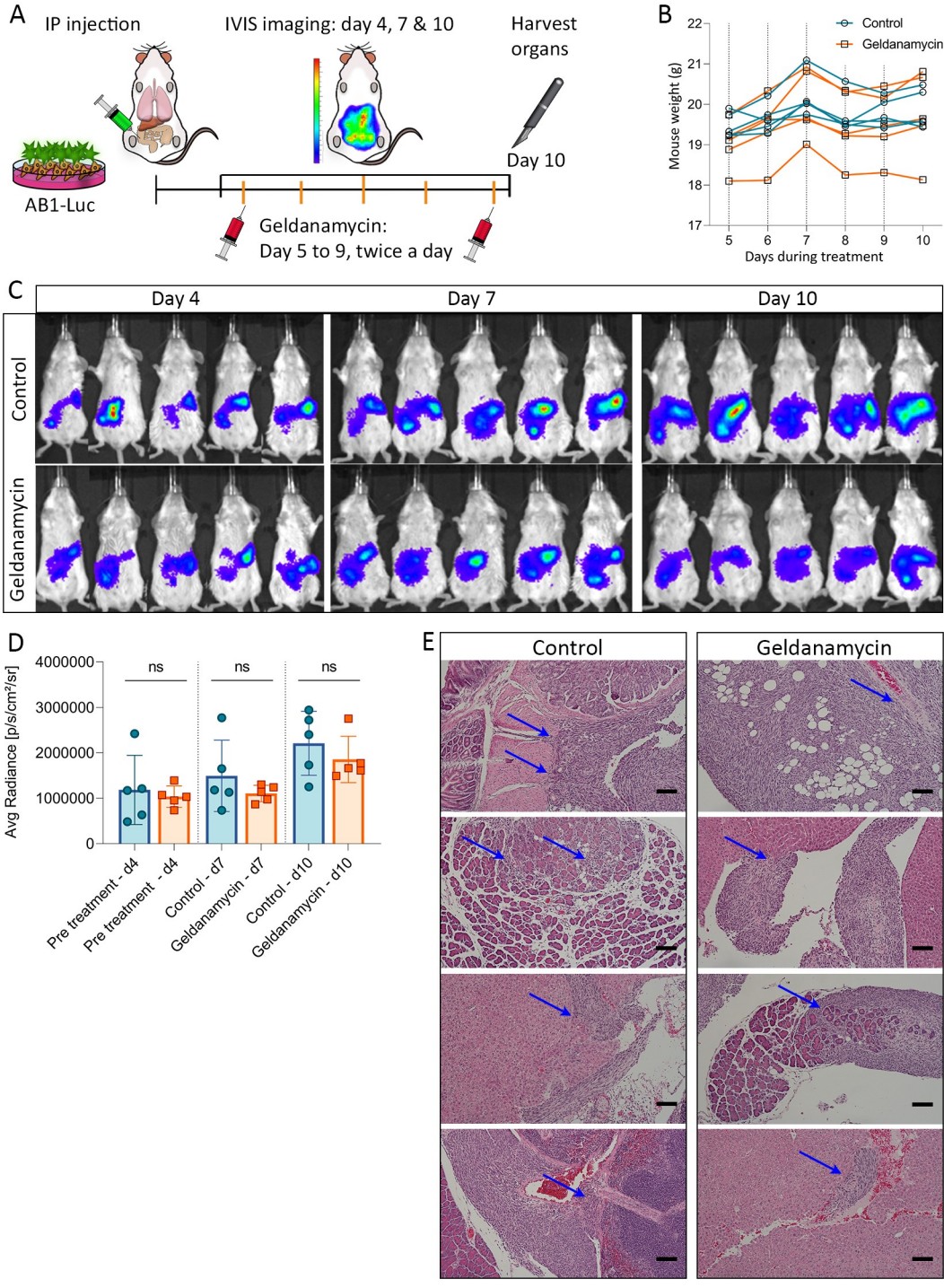

**Fig 3. Geldanamycin does not result in significant anti-mesothelioma activity *in vivo*.** (A) Experimental design. (B) Mice weight monitored daily during treatment. (C) Bioluminescence imaging for tumour bioluminescence monitoring on days 4, 7 and 10 for control group and geldanamycin group. (D) Tumour bioluminescence comparison based on average radiance (p/s/cm$^2$/sr) over time. Data are means ± SD values. n = 5. Unpaired t-test. (E) IP tumours were stained with hematoxylin and eosin. Blue arrows indicate tumour invasion in different organs. Scale bar = 100 μm.

This lack of translation from *in vitro* to *in vivo* activity could be related to an insufficiently wide therapeutic window, with tolerable drug exposure levels *in vivo* being too low to result in sufficient anti-cancer effect as we observed *in vitro*. In addition, it is possible that compensatory mechanisms are operational *in vivo* that allow the cancer cells to escape geldanamycin-induced anti-cancer effects. Alternatively, the myogenesis gene expression profile may be a bystander effect of exposure to the orthotopic environment, rather than a program driving invasive mesothelioma growth. Lastly, drug repurposing approaches have their limitations, in particular when based on an overlap in gene expression signatures between disease-associated tissues and cell lines treated with compounds, such as LINCS/CMap, resulting in false-positive hits [55–57]. Other treatment avenues will need to be explored to specifically target mesothelioma invasion.

## Methods

### Cell culture

Cell lines AE17 [58] and AB1 [59] were obtained from and verified by CellBank Australia. AB1 was transfected to express the luciferase (AB1-Luc) [60]. Cell lines MSTO-211H and NIH3T3 were obtained from and verified by ATCC. Cell lines VGE62, JU77 were established in-house from pleural mesothelioma patients [61]. AB1, AE17, VGE62, MSTO-211H, NIH3T3 and JU77 were maintained in complete R10 medium, RPMI 1640 (Invitrogen) supplemented with 10% foetal calf serum (FCS; Life Technologies), 20 mM HEPES (Sigma-Aldrich), 0.05 mM 2-mercaptoethanol (pH 7.2; Merck, Kilsyth, Australia), 60 μg/mL penicillin (Life Technologies), 50 μg/mL gentamicin (David Bull Labs). All cell lines were confirmed mycoplasma negative by polymerase chain reaction. All cells were cultured as a monolayer at 37 ˚C in a humidified atmosphere with 5% $CO_2$. Cells were passaged at approximately 1:10–1:5 every 2–3 days when reaching 75–80% confluence.

### Mice

This study was conducted in accordance with the institutional guidelines of the Harry Perkins Institute of Medical Research Animal Ethics Committee (approvals AE057 and AE183). Female BALB/c and Female C57BL/6 mice were obtained from the Animal Resource Centre (Murdoch, WA, Australia). All mice were 8 to 12 weeks of age when used for the experiments. Mice were housed at the Harry Perkins Institute of Medical Research Bioresources Facility under pathogen-free conditions at 21 ˚C to 22 ˚C with a 12/12 h light cycle. Cages (Tecniplast) had an individual air filtered system and contained aspen chips bedding (TAPVEI). Mice were fed Rat and Mouse cubes (Specialty Feeds) and had access to filtered water. Animal monitoring: mice were monitoring a minimum of twice weekly prior to the start of the experiment. Following ip tumour inoculation, clinical score and welfare points (S1 and S2 Tables) were closely followed to ensure animals were well. Mice were monitored the day after inoculation and daily monitored from day one of treatment (day 4 after inoculation) until end of experiment (day 10 after inoculation). Weight was recorded prior to first treatment injection and all mice were weighed daily during treatment. If weight dropped more than 10% mice stopped receiving treatment until recovered. Tumour size was measured using IVIS (See below: In vivo imaging system (IVIS)). The maximum tumour size reached in this study was measured in average radiance; 2.94 x $10^6$ photons/sec/cm$^2$/sr on day 10 after inoculation. For animal ethics purposes, endpoint was allowed on day 14 after inoculation, where the average radiance reaches 20 x $10^6$ photons/sec/cm$^2$/sr. This was based on pilot experiments demonstrating that there was no noticeable impost to the mice at that time point. For this study, the endpoint was selected based on treatment schedule (finishing on day 10) as opposed to waiting for clinical signs to appear.

## Tumour cell inoculation

Cells were trypsinized and washed two times in phosphate buffer saline (PBS) and counted with trypan blue dye. For the SC tumour model, mice were shaved on the right-hand flank and inoculated with $5 \times 10^5$ AB1 or $1 \times 10^6$ AE17 cells suspended in 100 μL PBS. Tumour volume (mm$^3$) was monitored with callipers. For the IPL tumour model, mice were anaesthetised under continuous isofluorane and inoculated with $5 \times 10^5$ AB1 or $1 \times 10^6$ AE17 cells suspended in 200 μL PBS into the pleural space as previously described [62]. For the IP tumour model, mice where inoculated in the intraperitoneal cavity on the right flank with $5 \times 10^5$ AB1-Luc cells suspended in 200 μL PBS. Tumour size was determined by in vivo imaging system, see below. All mice were euthanized in accordance with animal ethics guidelines.

## Tumour preparation for RNA sequencing

At day 10 after tumour inoculation, mice were euthanized and the tumours were harvested and submerged in RNAlater (Life Technologies) at 4 ˚C overnight to allow RNAlater to penetrate the tissue. Then, tumours were removed from RNAlater and stored at –80 ˚C until dissociation with TRIzol (Life Technologies) using a TissueRuptor (QIAGEN). RNA was extracted with chloroform and purified on RNeasy MinElute columns (QIAGEN).

## RNA sequencing analysis

Library preparation and sequencing at 50 bp single end reads with Illumina HiSeq standard protocols were performed by Australian Genome Research Facility.

Alignment was performed using Kallisto [63]. Differentially expressed genes were identified between IPL and SC tumours within both AB1 and AE17 models using DESeq2 [15]. P values were adjusted for multiple comparisons using the Benjamini–Hochberg (B-H) method. A p value < 0.05 was considered significant. Differentially expressed genes were analysed as follows: Ingenuity Systems [16] was used to identify predicted upstream regulators, using right-tailed Fisher's exact tests and default settings for other options; activation Z-scores were calculated for each regulator by comparing their known effect on downstream targets with observed changes in gene expression. Those with activation Z-scores ≥2 or ≤2 were considered "activated" or "inhibited", respectively. Pathways analysis from InnateDB [19, 64] was used to identify relevant canonical pathways. Enrichment analysis was performed with Enrichr [24–26]. LINCS L1000 Chem Pert repository [22] and CMap [11] were used to identify drugs that were predicted to phenocopy our gene expression profile of interest (P value < 0.05). Pre-ranked GSEA [20, 21] was used to identify enriched gene sets from the gene ontology [22] and hallmark [23] consortiums; p. value < 0.05 and FDR < 0.25 were considered significant.

## Soft agar colony formation

Cells were incubated with different concentrations of geldanamycin (100, 50, 25, 12.5 6.25 and 0 nM) for a soft agar colony formation assay [65]. Briefly, 3% 2-hydroxyethyl agarose (agarose, A4018; Sigma-Aldrich, Australia) was prepared as stock solution. The bottom layer was prepared by incubating 20% agarose gel in R10 complete medium, RPMI 1640 (Invitrogen) supplemented with 10% foetal calf serum (FCS; Life Technologies), 20 mM HEPES (Sigma-Aldrich), 0.05 mM 2-mercaptoethanol (pH 7.2; Merck, Kilsyth, Australia), 60 μg/mL penicillin (Life Technologies), 50 μg/mL gentamicin (David Bull Labs), for a final concentration of 0.6%, 2mL per well, at 4 ˚C for 1 h to allow the mixture to solidify, then incubating at 37 ˚C for at least 30 min before seeding the cells. The cell-containing layer (10% agarose gel in R10 complete medium for a final concentration of 0.3%, 1 mL per well) was prepared with a concentration of

1,000 cells/mL. 1 mL of cells was then transfer to each well. The feeder layer (10% agarose gel in R10 complete medium for a final concentration of 0.3%, 1 mL per well was prepared with different concentrations of geldanamycin; each different concentration was then added to each well on top of the cells. The 6-well plate was then incubated at 4 ˚C for 15 min before incubating at 37 ˚C with 5% $CO_2$ for a week. A new feeder layer was added once per week until day 22.

Colony counting was performed by adding 1 mL of 0.005% crystal violet (C0775; Sigma-Aldrich, Australia) in PBS on top of each well and incubating at room temperature for 24 h. Pictures were taken and colonies counted with ImageJ (v1.52a).

## Migration and matrigel invasion assay

Cells were harvested and seeded in Incucyte ImageLock 96-well plates (Essen BioScience) at a density of 10 x $10^4$ cells/mL overnight. A scratch wound was performed on the confluent cells with the 96-pin IncuCyte WoundMaker Tool (Essen BioScience). For migration assays, cells were washed with PBS once before adding 100 µL media with geldanamycin at indicated concentrations. For matrigel invasion assays, cells were washed with PBS once before adding 50 µL of matrigel basement membrane matrix (FAL356231; Corning, NY, USA) at 8 mg/mL. Lastly, 100 µL media with geldanamycin at indicated concentrations were added on top of the matrigel. Cells were incubated in the IncuCyte ZOOM at 37 ˚C with 5% $CO_2$ and pictures were taken every 2 hours. Data were analysed using the IncuCyte Scratch Wound Cell Migration Software Module, calculating the Relative Wound Density (%) (RWD) for both migration and invasion assays.

## MTT assay

Cells were harvested and seeded in 96-flat well plates (Corning) at a density of 5 x $10^4$ cells/mL overnight. Media was removed and 100 µL media with geldanamycin at indicated concentrations was added to each well. Cell viability and toxicity were measured at 48 h. Cells were incubated with 50 µL of a 2 mg/mL solution of (3-(4,5-dimethyl-thiazole-2-yl)-2,5-biphenyl tetrazolium (MTT, Sigma-Aldrich) in PBS for 4 h and then exposed to 100 µL dimethyl sulfoxide (DMSO; Sigma-Aldrich). Cell viability was measured by absorption at 570 nm in a microplate spectrophotometer (Spectromax 250 plate reader). Results are shown as relative cell viability. MTT obtained values were normalized and concentrations transformed to logarithm, the nonlinear regression (curve fit) and $IC_{50}$ were calculated using a log(geldanamycin) versus response algorithm.

## Flow cytometry

Cells were passaged two to three times prior flow cytometry. Cells were Trypsinised and plated in a Cells were plated in a 96-well rounded bottom plate and washed with PBS. Antibody information is summarised in S3 Table. Fc Block was added to all wells and plate was incubated at 4 ˚C for 30 min. Cells were washed once with PBS and viability antibody was added in PBS for 20 min at room temperature. Cells were washed once with PBS and surface antibody cocktail, or single stains or FMOs, were added to each well in PBS + 2% FBS buffer. Plate was incubated at 4 ˚C for 30 min and cells washed twice with PBS + 2% NCS buffer. Data were recorded with the cytometer LSR Fortessa and analysed using FlowJo Software.

## Protein extraction and quantification

Protein extraction for western blotting with RIPA buffer was performed following the manufacturer guideline's (Thermo Fisher). RIPA buffer was supplemented with PhosSTOP inhibitor

(Merck) at 10X concentration and cOmplete EDTA-free Proteinase Inhibitor Cocktail (Roche) at 25X concentration. Trypsinised cells were washed twice with PBS and cold complete RIPA buffer was added to each vessel and kept on ice for 5 min. The lysate was collected with a scrapper and centrifuged at 4 ˚C at maximum speed for 15 min. The supernatant was collected and quantified with the Pierce BCA Protein Assay Kit (Thermo Fisher) as explained by the manufacturer guidelines. The standard curve was prepared in triplicate with albumin standard (BSA) diluted in PBS, working concentrations are 2, 1.5, 1, 0.5, 0.2 and 0 μg/μL. The BCA working reagent was prepared by mixing 50 parts of BCA Reagent A with 1 part of BCA reagent B. All the standards and samples were prepared by triplicate, and 10 μL of each standard or sample were added to a 96-well plate with 200 μL of BCA working reagent and mixed by pipetting. The plate was incubated for 30 min at 37 ˚C. Absorbance was measured at 562 nm on a plate reader and the standard curve was used to determine the protein concentration of each sample.

## Western blotting

Protein lysates were thaw and diluted to a final concentration of 30 μg of protein and combined with 4X loading buffer (4X Laemmli Sample Buffer completed with 2-mercaptoethanol, Bio-Rad) for a loading volume of 30 μL. The lysates were incubated 5 min at 95 ˚C. 10% Mini-Protean TGX Pre-cast Protein Gel (Bio-Rad) were used to load the ladder (2 μL, Precision Plus Protein Dual Colour Standards, Bio-Rad), followed by 30 μL of each sample. 1X Western Running Buffer was added to cover the gel (10X Western Running Buffer (SDS-PAGE), 250 mM Tris, 1.92 M glycine, 1% SDS, pH 8.3). Gels ran for 25 min at 200 V.

Using the Trans-blot Turbo (Bio-Rad) for 7 min, the gel was then transferred to a 0.2 μm membrane (Trans-blot Turbo Mini Nitrocellulose Transfer Pack, Bio-Rad). Once transferred, the membrane was blocked with 5% non-fat milk powder in 1X TBST Buffer (Tris-Buffer Saline with Tween, 20 mM Tris, 500 mM NaCl, and 0.05% Tween 20) at room temperature for 60 min, followed by three washes for 5 min in 1X TBST. The primary antibody, GAPDH (1:1000) (GAPDH (14C10) Rabbit mAb, Cell Signaling, 2118S), or HSP90 (1:1000) (HSP90 (C45G5) Rabbit mAb, Cell Signaling, 4877S) was diluted in 5% BSA in 1X TBST and added to the membrane for an overnight incubation at 4 ˚C. The membrane was washed three times with 1X TBST and the secondary antibody, Anti-rabbit IgG, HRP-linked Antibody (1:2000) (Cell Signaling, 7074P2), was diluted in 5% non-fat milk powder in 1X TBST for 90 min at room temperature, followed by three washes for 5 min with 1X TBST. The membrane was placed in the ChemiDoc. To visualise the target protein, Clarity Western ECL Substrate (Bio-Rad) was prepared as per manufacturer guidelines at 1:1 ratio and added on top of the membrane to cover the surface for 5 min at room temperature. ECL excess was drained, and the membrane placed in the ChemiDoc to visualise the target protein.

## In vivo geldanamycin treatment

Female BALB/cJAusb mice (10–12 weeks) were inoculated IP with 5 x $10^5$ AB1-Luc on their right flank. Mice were randomly allocated to the different groups on the first treatment day. Initial mouse weight was measured immediately before the first geldanamycin injection, and it was used to calculate the amount to dose (5 mL/kg). Treatment with geldanamycin started on day 5 after tumour inoculation. Geldanamycin (1, 0.5 or 0.1 mg/kg) was prepared in sterile DMSO 1% and saline and dosed to all mice via a single intraperitoneal injection. Mice were dosed for a maximum of five consecutive days, twice a day (8 hours apart). In vivo imaging system (IVIS) (see below) was performed on days 4 (before treatment), 7 (during treatment) and 10 (after treatment). Mice were euthanized on day 10 once bioluminescence image was

completed and in accordance with animal ethics guidelines and organs were harvested for staining.

## In vivo imaging system (IVIS)

XenoLight D-Luciferin potassium salt (PerkinElmer, VIC, Australia) was used at a 150 mg/kg concentration dissolved in sterile PBS. Approximately 150 μL (15 mg/mL concentration) was injected subcutaneously per mouse. Mice were anaesthetised in a chamber with a controlled flow of isoflurane 2% and oxygen flow rate of 1 L/min. When mice were fully unconscious, eye gel was applied to moisturise eyes during the imaging process. Mice were then transferred to the IVIS Lumina II camera chamber and isofluorane was decreased to 0.5–1% and oxygen flow rate to 0.8 L/min. Mice were imaged for 5 second exposure duration at 13 min post injection. Tumour burden is calculated as average radiance (photons/sec/cm$^2$/sr).

## Hematoxylin and eosin staining

Mouse tissues were fixed with 4% paraformaldehyde for 48–72 hours and embedded in paraffin. 5 μm sections were cut by microtome (Thermo Fisher Scientific) and placed on Microscope Slides with 20 mm Colourfrost blue (Hurst Scientific). Slides were deparaffinised at room temperature as follows: 2 rounds of xylene (3 min each), 2 rounds of 100% ethanol (2 min each), 95% ethanol (1 min), 70% ethanol (1 min), 40% ethanol (1 min), 3 rounds of distilled H$_2$O (3 min each). Slides were stained with Mayers hematoxylin (Sigma-Aldrich) for 10 min and rinsed with running tap water, the counterstain was performed with acidified Eosin Y solution (Sigma-Aldrich) (0.5% glacial acetic acid) for 45 seconds. Dehydration was performed as follows: 40% ethanol (30 sec), 70% ethanol (30 sec), 95% ethanol (30 sec), 2 rounds of 100% ethanol (1 min each), 2 rounds of xylene (3 min each). Mounting was performed with Pertex mounting medium (Histolab), and sections were imaged under a light microscope.

## Quantification and statistical analysis

GraphPad Prism software was used to determine statistical significance of differences between groups by unpaired t-test when comparing two groups. A p value < 0.05 was considered significant. Each figure legend contains all the statistical details on each experiment, including the specific statistical test for that assay, exact value of n, what n represents and dispersion and precision measures. RNA sequencing statistical details can be found under Methods: RNA sequencing analysis.

## Supporting information

**S1 Fig. Mesothelial cell signature is associated with the pleural space.** (A) Enrichment analysis of IPL (log fold change >1) differentially expressed genes using the Descartes dataset in Enrichr, p values 0.05 and 0.01 are represented with dotted line on–log10 = 1.3 and 2, respectively.
(TIF)

**S2 Fig. Luciferase transduction does not affect the AB1 cell line phenotype.** (A–C) Flow cytometry for surface markers (A) PD-L1, (B) MHC-I, and (C) Podoplanin in AB1 and AB1-Luc cell lines. Data is represented as MFI (Median). N = 5. (D) MTT assay in AB1 and AB1-Luc cell lines. The curve is presented as a non-linear regression; log(geldanamycin) versus response. IC$_{50}$ values are in nM. Data are presented as means ± SD of 3 replicates. (E–G) Scratch assay for migration in (E) AB1 and (F) AB1-Luc cell lines. Geldanamycin was serial diluted from 400 nM to 1.25 nM, with 0 nM as control. (G) AB1 and AB1-Luc side by side

comparison. (H) Western blot for cell lines AB1 and AB1-Luc shows housekeeping protein GAPDH at 37 kDa, and HSP90 protein at 90 kDa.
(TIF)

**S3 Fig. Geldanamycin causes weight loss when dosed at 1 or 0.5 mg/kg *in vivo*.** (A) Mice weight monitored daily during treatment at 1 mg/kg, each dotted line indicates the number of doses. (B) Mice weight monitored daily during treatment at 0.5 mg/kg, each doted like indicates two doses on that specific day.
(TIF)

**S1 Table. Mice clinical signs scoring criteria.**
(DOCX)

**S2 Table. Mice intervention criteria.**
(DOCX)

**S3 Table. Flow cytometry panel antibody information.**
(DOCX)

**S1 File. LINCS and CMap results.** AB1 and AE17 pleural differentially expressed genes.
(XLSX)

## Acknowledgments

The authors acknowledge the facilities and scientific and technical assistance of the National Imaging Facility, a National Collaborative Research Infrastructure Strategy (NCRIS) capability, and Microscopy Australia at the Centre for Microscopy, Characterisation and Analysis, The University of Western Australia.

## Author Contributions

**Conceptualization:** Y. C. Gary Lee, Richard A. Lake, W. Joost Lesterhuis.

**Data curation:** M. Lizeth Orozco Morales, Emma de Jong.

**Formal analysis:** M. Lizeth Orozco Morales, Emma de Jong, Rachael M. Zemek.

**Funding acquisition:** Sally M. Lansley, Y. C. Gary Lee, Anthony Bosco, Richard A. Lake, W. Joost Lesterhuis.

**Investigation:** M. Lizeth Orozco Morales, Catherine A. Rinaldi.

**Methodology:** M. Lizeth Orozco Morales, Catherine A. Rinaldi, Emma de Jong, Sally M. Lansley, Rachael M. Zemek, Anthony Bosco.

**Project administration:** W. Joost Lesterhuis.

**Resources:** W. Joost Lesterhuis.

**Software:** Emma de Jong, Anthony Bosco.

**Supervision:** Richard A. Lake, W. Joost Lesterhuis.

**Validation:** M. Lizeth Orozco Morales.

**Visualization:** M. Lizeth Orozco Morales.

**Writing – original draft:** M. Lizeth Orozco Morales, Richard A. Lake, W. Joost Lesterhuis.

**Writing – review & editing:** M. Lizeth Orozco Morales, Catherine A. Rinaldi, Emma de Jong, Sally M. Lansley, Y. C. Gary Lee, Rachael M. Zemek, Anthony Bosco, Richard A. Lake, W. Joost Lesterhuis.

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
