## [Decision Letter · Decision Letter 0]

21 Nov 2022

PONE-D-22-23207Geldanamycin treatment does not result in anti-cancer activity in a preclinical model of orthotopic mesotheliomaPLOS ONE

Dear Dr. Orozco Morales,

Thank you for submitting your manuscript to PLOS ONE. After careful consideration, we feel that it has merit but does not fully meet PLOS ONE’s publication criteria as it currently stands. Therefore, we invite you to submit a revised version of the manuscript that addresses the points raised during the review process.

Please read carefully the major issues raised by both reviewers, and try to implement them in a revised version.It has potential to be a very interesting and good manuscript, so I hope you will be able and willing to proceed with revision.

We look forward to receiving your revised manuscript.

Kind regards,

Luka Brcic

Academic Editor

PLOS ONE

Journal Requirements:

“Funding for this work was received from Cancel Council WA and iCare Dust Diseases Care. W.J.L. was funded by fellowships from the NHMRC, the Simon Lee Foundation and Cancer Council of Western Australia. M.L.O.M acknowledges the Commonwealth Government’s support through the University Postgraduate Award and Australian Government Research Training Program Scholarships at The University of Western Australia. The authors also acknowledge the facilities and scientific and technical assistance of the National Imaging Facility, a National Collaborative Research Infrastructure Strategy (NCRIS) capability, and Microscopy Australia at the Centre for Microscopy, Characterisation and Analysis, The University of Western Australia. A facility funded by the University, State and Commonwealth Governments.”

“This research was funded by Cancel Council WA and iCare Dust Diseases Care. W.J.L. was funded by fellowships from the National Health and Medical Research Council's (NHMRC), the Simon Lee Foundation and Cancer Council of Western Australia. The funders had no role in study design, data collection and analysis, decision to publish, or preparation of the manuscript.

Cancer Council WA: https://cancerwa.asn.au/

iCare: https://www.icare.nsw.gov.au/

NHMRC: https://www.nhmrc.gov.au/

Simon Lee Foundation: https://simonleefoundation.com.au/”

Reviewers' comments:

Reviewer's Responses to Questions

**Comments to the Author**

1. Is the manuscript technically sound, and do the data support the conclusions?

Reviewer #1: Partly

Reviewer #2: Yes

2. Has the statistical analysis been performed appropriately and rigorously? 

Reviewer #1: Yes

Reviewer #2: Yes

3. Have the authors made all data underlying the findings in their manuscript fully available?

Reviewer #1: Yes

Reviewer #2: Yes

4. Is the manuscript presented in an intelligible fashion and written in standard English?

Reviewer #1: Yes

Reviewer #2: Yes

5. Review Comments to the Author

Reviewer #1: Morales et al investigate geldanamycin as a treatment for mesothelioma. They implant two murine mesothelioma cells subcutaneously and orthotopically into mice and identify differentially expressed genes between the invasive growth in the orthotopic location and the non-invasive growth of the sucutaneously implanted tumors. They perform pathway and database analyses with the differentially expressed genes to identify a gene expression signature associated with invasive growth and geldanamycin as a potential antagonist of this signature. They subsequently test geldanamycin in vitro and in vivo and find reduction of cell growth, migration and invasion in vitro but no inhibition of tumor growth or invasion in vivo.

The experiments appear technically well performed and the manuscript is well written.

I have two main concerns:

1) The difference in growth characteristics between the models is dictated by the location and may thus not reflect the aggressiveness of the tumor cells. I would be more interested in a difference between cell models that do and do not grow invasively in the pleura.

2) The authors do not explore the difference between their in vitro and in vivo findings. According to the literature HSP90 is the target of geldanamycin. Is this target present in their models? How is it affected by geldanamycin treatment in vitro and in vivo? Have others found efficacy of geldanamycin in vivo in other cancers?

Reviewer #2: The Authors compared an invasive pleural model with a non-invasive

subcutaneous model of mesothelioma. They performed transcriptomic analyses on the tumour samples and used differentially expressed genes between pleural and subcutaneous mesothelioma to interrogate platforms allowing to identify drugs able to produce a similar transcriptional difference. Within the list of drugs, the Authors select geldanamycin, an hsp90 inhibitor. This drug has already been shown to inhibit mesothelioma cell growth in vitro and the Authors confirm that nanomolar concentrations of geldanamycin inhibit cancer cell growth, invasion, and migration in vitro in the four mesothelioma cell lines tested. However, they observed no effect in vivo on growth or invasion in an orthopic model using cells labeled with a luciferase reporter.

This study includes some interesting observations, however, several issues have to be addressed before this manuscript can be considered for publication.

Major:

1. If the Authors select log fold change >1, and exclude Ig genes, where differential expression might result from the tumor microenvironment, the remaining 99 genes allow to identify “mesothelial cell signature” in the enrichr plateform that they used. This means that the pleural environment is more favorable to mesothelial differentiation program commitment. This is an important information that the Authors should add for the benefit of the community. It also provides a slightly different perspective compared to what the Authors have proposed as data “suggesting that muscle development signatures were related to the invasive phenotype of pleural mesothelioma”. Indeed, it adds more emphasis on the fact that orthotopic environment favors the maintenance of mesothelial differentiation. This does not preclude the selection of geldanamycin for further investigations.

2. The Authors should provide evidence whether the AB1-luciferase cell line used in the orthotopic model has the same profile as parental cells. For example they could test whether they maintain the high expression of UPK3B and LRRN4 as parental cells.

3. The Authors could compare whether AB1 and AB1-luc show the same sensitivity to geldanamycin growth inhibition effects.

4. The Authors could also test whether there are different levels of HSP90 and a different ratio of HSP90 alpha vs beta isoforms between AB1 cells used for the transcriptomic analysis and the AB1-luciferase cells, which may help explain difference in sensitivity.

6. PLOS authors have the option to publish the peer review history of their article (what does this mean?). If published, this will include your full peer review and any attached files.

Reviewer #1: No

Reviewer #2: **Yes: **Emanuela Felley-Bosco

---

## [Author Response · Author response to Decision Letter 0]

21 Mar 2023

PONE-D-22-23207R1

Geldanamycin treatment does not result in anti-cancer activity in a preclinical model of orthotopic mesothelioma

Orozco Morales et al. 

Point-by-point reply to reviewers’ comments. 

We thank the reviewers for their constructive comments, which have helped us to improve our manuscript. Here, we present a point-by-point reply addressing all comments.

Reviewer #1: 

Morales et al investigate geldanamycin as a treatment for mesothelioma. They implant two murine mesothelioma cells subcutaneously and orthotopically into mice and identify differentially expressed genes between the invasive growth in the orthotopic location and the non-invasive growth of the sucutaneously implanted tumors. They perform pathway and database analyses with the differentially expressed genes to identify a gene expression signature associated with invasive growth and geldanamycin as a potential antagonist of this signature. They subsequently test geldanamycin in vitro and in vivo and find reduction of cell growth, migration and invasion in vitro but no inhibition of tumor growth or invasion in vivo.

The experiments appear technically well performed and the manuscript is well written.

I have two main concerns:

1) The difference in growth characteristics between the models is dictated by the location and may thus not reflect the aggressiveness of the tumor cells. I would be more interested in a difference between cell models that do and do not grow invasively in the pleura.

We agree with the reviewer that having a cell line that does not grow invasively in the pleural cavity would be of great interest, but the two different cell lines that we tested (AB1 and AE17), in two different mouse strains (C57BL/6 and BALB/c), showed invasive and fast-proliferative behaviour in the pleural space, but not in the subcutaneous space, which is why we compared the transcriptomes of the tumours derived from the same cell line, yet growing in different anatomical compartments.

2) The authors do not explore the difference between their in vitro and in vivo findings. According to the literature HSP90 is the target of geldanamycin. Is this target present in their models? How is it affected by geldanamycin treatment in vitro and in vivo? Have others found efficacy of geldanamycin in vivo in other cancers?

We chose geldanamycin using the LINCS drug repurposing database as it was predicted to antagonise the invasion-associated gene expression profile. The source data in the LINCS database are derived from cell lines treated with thousands of compounds. The gene expression profile induced by geldanamycin can be HSP90-independent. Indeed, geldanamycin was initially discovered as an antibiotic, and only decades later was to be shown to bind to HSP90. However, to address the presence of HSP90 present in our models and to observe if geldanamycin downregulates HSP90 in vitro, we now added western blots in Supplementary figure 2 (Fig S2) from both cell lines that were used for in vitro and in vivo experiments, AB1 and AB1-Luc respectively. These data show the same basal expression of HSP90 in both cell lines, as well as an unchanged expression after addition of geldanamycin at 500 nM, a concentration which in our migration experiments reduced cell migration. These data agree with others that have shown that geldanamycin treatment does not downregulate HSP90 protein expression levels in urinary bladder cancer cell lines RT4 and T24 (Karkoulis et al., 2013), and breast cancer cell line MCF-7 (Shin et al., 2020), but rather inhibits HSP90’s downstream targets (Karkoulis et al., 2013) (Shin et al., 2020), suggesting that geldanamycin’s mechanism of action is either HSP90-independent or works through inhibition of its downstream targets as per the literature. 

We have added the following sentences to the Discussion section (page 8, line 164 - 169): “In our study, we observed that addition of geldanamycin did not affect the expression of HSP90 itself. These data agree with others that have shown that geldanamycin treatment does not downregulate HSP90 protein expression levels as such (49) (50), but rather inhibits HSP90’s downstream targets (49) (50). Another explanation for geldanamycin’s mechanism of action as observed in our in vitro experiments is that it is HSP90-independent.”

Supplementary Fig S2. Luciferase transduction does not affect the AB1 cell line phenotype (H) Western blot for cell lines AB1 and AB1-Luc shows housekeeping protein GAPDH at 37 kDa, and HSP90 protein at 90 kDa.

Reviewer #2: 

The Authors compared an invasive pleural model with a non-invasive subcutaneous model of mesothelioma. They performed transcriptomic analyses on the tumour samples and used differentially expressed genes between pleural and subcutaneous mesothelioma to interrogate platforms allowing to identify drugs able to produce a similar transcriptional difference. Within the list of drugs, the Authors select geldanamycin, an hsp90 inhibitor. This drug has already been shown to inhibit mesothelioma cell growth in vitro and the Authors confirm that nanomolar concentrations of geldanamycin inhibit cancer cell growth, invasion, and migration in vitro in the four mesothelioma cell lines tested. However, they observed no effect in vivo on growth or invasion in an orthopic model using cells labeled with a luciferase reporter.

This study includes some interesting observations, however, several issues have to be addressed before this manuscript can be considered for publication.

Major:

1. If the Authors select log fold change >1, and exclude Ig genes, where differential expression might result from the tumor microenvironment, the remaining 99 genes allow to identify “mesothelial cell signature” in the enrichr plateform that they used. This means that the pleural environment is more favorable to mesothelial differentiation program commitment. This is an important information that the Authors should add for the benefit of the community. It also provides a slightly different perspective compared to what the Authors have proposed as data “suggesting that muscle development signatures were related to the invasive phenotype of pleural mesothelioma”. Indeed, it adds more emphasis on the fact that orthotopic environment favors the maintenance of mesothelial differentiation. This does not preclude the selection of geldanamycin for further investigations.

We thank the reviewer for this observation. We have performed this analysis ourselves and have indeed observed a mesothelial cell signature for different cell types. We added the data as Supplementary figure 1 (Fig S1) and discussed in the Results section (page 5, line 71 - 75): “Further analysis with the Descartes dataset in Enrichr (24) (25) (26) of the IPL differentially expressed genes (log fold change >1) while excluding immunoglobulin-related genes resulted in a mesothelial cell signature (Fig S1A), suggesting that environmental cues from the pleural space drive this mesothelial program more than the subcutaneous space”.

Figure S1A. Enrichment analysis of IPL (log fold change >1) differentially expressed genes using the Descartes dataset in Enrichr, p values 0.05 and 0.01 are represented with dotted line on –log10 = 1.3 and 2, respectively.

2. The Authors should provide evidence whether the AB1-luciferase cell line used in the orthotopic model has the same profile as parental cells. For example they could test whether they maintain the high expression of UPK3B and LRRN4 as parental cells.

3. The Authors could compare whether AB1 and AB1-luc show the same sensitivity to geldanamycin growth inhibition effects.

To address questions 2 and 3 from Reviewer #2, we looked at the expression of the surface markers PD-L1, MHC-I and podoplanin with flow cytometry, and performed in vitro scratch migration and cytotoxicity assays with geldanamycin in both cell lines. We did not observe any difference between AB1 and AB1-Luc. 

These results in Supplementary figure 1 (Fig S2) under the section: Geldanamycin treatment does not have anti-mesothelioma activity in vivo (Fig 3), in page 7, lines 120 – 125: “To assess any therapeutic effect of geldanamycin on mesothelioma invasion and growth in vivo, we used our optimised model of orthotopic invasive mesothelioma (14) with AB1 cells expressing luciferase (AB1-Luc). The luciferase transduction had not changed the phenotype and geldanamycin sensitivity of the AB1 cell line, in terms of its expression of PD-L1, MHC-I and podoplanin, nor its in vitro invasive potential (Fig S2A-G). In addition, the basal levels of HSP90 did not change between cell lines, with or without geldanamycin (Fig S2H).”

Supplementary Fig S2. Luciferase transduction does not affect the AB1 cell line phenotype (A - C) Flow cytometry for surface markers (A) PD-L1, (B) MHC-I, and (C) Podoplanin in AB1 and AB1-Luc cell lines. Data is represented as MFI (Median). N = 5. (D) MTT assay in AB1 and AB1-Luc cell lines. The curve is presented as a non-linear regression; log(geldanamycin) versus response. IC50 values are in nM. Data are presented as means ± SD of 3 replicates. (E - G) Scratch assay for migration in (E) AB1 and (F) AB1-Luc cell lines. Geldanamycin was serial diluted from 400 nM to 1.25 nM, with 0 nM as control. (G) AB1 and AB1-Luc side by side comparison. (H) Western blot for cell lines AB1 and AB1-Luc shows housekeeping protein GAPDH at 37 kDa, and HSP90 protein at 90 kDa.

4. The Authors could also test whether there are different levels of HSP90 and a different ratio of HSP90 alpha vs beta isoforms between AB1 cells used for the transcriptomic analysis and the AB1-luciferase cells, which may help explain difference in sensitivity.

To address question 4, as previously mentioned in question 2 from Reviewer #1, we performed western blot on both cell lines to determine the basal levels of HSP90 and after geldanamycin treatment at 500 nM. These results are similar to what others have found when using geldanamycin at similar concentrations in vitro. This could indicate that geldanamycin does not directly downregulate HSP90 and instead reduces is action, affecting its downstream targets. These results are now discussed in the Discussion section (page 8, line 164 - 169) (see response to question 2, Reviewer #1).

---

## [Editor Report · Decision Letter 1]

27 Mar 2023

Geldanamycin treatment does not result in anti-cancer activity in a preclinical model of orthotopic mesothelioma

PONE-D-22-23207R1

Dear Dr. Orozco Morales,

We’re pleased to inform you that your manuscript has been judged scientifically suitable for publication and will be formally accepted for publication once it meets all outstanding technical requirements.

Kind regards,

Luka Brcic

Academic Editor

PLOS ONE

Additional Editor Comments (optional):

Thank you fro additional analyses and changes according to reviewers´ comments, which improved the quality and clarity of your manuscript.
---

## [Editor Report · Acceptance letter]

25 Apr 2023

PONE-D-22-23207R1 

Geldanamycin treatment does not result in anti-cancer activity in a preclinical model of orthotopic mesothelioma 

Dear Dr. Lesterhuis:

I'm pleased to inform you that your manuscript has been deemed suitable for publication in PLOS ONE. Congratulations! Your manuscript is now with our production department. 

Kind regards, 

on behalf of

Dr. Luka Brcic 

Academic Editor

PLOS ONE